

# Current practice in the management of new-onset atrial fibrillation in critically ill patients: a UK-wide survey

Chung Shen Chean[1], Daniel McAuley[2], Anthony Gordon[3] and Ingeborg Dorothea Welters[1,4]

[1] Intensive Care Unit, Royal Liverpool University Hospital, Liverpool, United Kingdom
[2] School of Medicine, Dentistry and Biomedical Sciences, The Queen's University Belfast, Belfast, United Kingdom
[3] Faculty of Medicine, Department of Surgery & Cancer, Section of Anaesthetics, Pain Medicine and Intensive Care, Imperial College London, London, United Kingdom
[4] Institute of Ageing and Chronic Disease, University of Liverpool, Liverpool, United Kingdom

Corresponding author
Ingeborg Dorothea Welters, I.Welters@liverpool.ac.uk

## ABSTRACT

**Background.** New-onset atrial fibrillation (AF) is the most common arrhythmia in critically ill patients. Although evidence base and expert consensus opinion for management have been summarised in several international guidelines, no specific considerations for critically ill patients have been included. We aimed to establish current practice of management of critically ill patients with new-onset AF.

**Methods.** We designed a short user-friendly online questionnaire. All members of the Intensive Care Society were invited via email containing a link to the questionnaire, which comprised 21 questions. The online survey was conducted between November 2016 and December 2016.

**Results.** The response rate was 397/3152 (12.6%). The majority of respondents (81.1%) worked in mixed Intensive Care Units and were consultants (71.8%). Most respondents (39.5%) would start intervention on patients with fast new-onset AF and stable blood pressure at a heart rate between 120 and 139 beats/min. However, 34.8% of participants would treat all patients who developed new-onset fast AF. Amiodarone and beta-blockers (80.9% and 11.6% of answers) were the most commonly used anti-arrhythmics. A total of 63.8% of respondents do not regularly anti-coagulate critically ill patients with new-onset fast AF, while 30.8% anti-coagulate within 72 hours. A total of 68.0% of survey respondents do not routinely use stroke risk scores in critically ill patients with new-onset AF. A total of 85.4% of participants would consider taking part in a clinical trial investigating treatment of new-onset fast AF in the critically ill.

**Discussion.** Our results suggest a considerable disparity between contemporary practice of management of new-onset AF in critical illness and treatment recommendations for the general patient population suffering from AF, particularly with regard to anti-arrhythmics and anti-coagulation used. Amongst intensivists, there is a substantial interest in research for management of new-onset AF in critically ill patients.

## INTRODUCTION

Atrial fibrillation (AF) is the most common cardiac arrhythmia in both the general population and the critical care setting (*Makrygiannis et al., 2014*; *Seguin et al., 2004*). New-onset fast AF is defined as atrial fibrillation with a rapid ventricular response of more than 100 beats per minute (bpm) in patients without a previous history of atrial fibrillation.

In the past decades there has been increased attention to new-onset AF in patients in critical care, because it is associated with a worse prognosis (*Chen et al., 2015*; *Shaver et al., 2015*). Evidence is growing that new-onset AF is associated with longer Intensive Care Unit (ICU) stay and higher mortality (*Reinelt et al., 2001*; *Tseng et al., 2016*; *Yoshida et al., 2015*). Observational studies suggested that the prevalence of AF in non-cardiac medical ICUs ranges from 5 to 26% (*Carrera et al., 2016*; *Chen et al., 2015*), and affects up to 10% of patients in surgical ICUs (*Knotzer et al., 2000*; *Seguin et al., 2004*).

However, despite the relatively high incidence of new-onset AF among critically ill patients, there is paucity of evidence for its management in the critical care setting. In particular, little is known about new-onset AF in comparison to pre-existing AF in this subset of patients. Previous studies have mainly investigated the epidemiology, risk factors and outcomes of new-onset atrial fibrillation in critical care, while little evidence, which is mainly based on small single-centre studies, is available for treatment (*Yoshida et al., 2015*).

As a result, guidance on management of AF is based on the evidence obtained in the general population and does not include patients in intensive care. We performed an online survey amongst practising intensivists in the United Kingdom to establish current practice regarding the management of critically ill patients developing new-onset AF. We also aimed to identify areas in which adherence to current guidance is low and further research is needed to clarify uncertainties in the treatment of these patients.

## METHODS

We designed a short user-friendly online questionnaire using Qualtrics® (Provo, UT, USA) to assess the knowledge and to explore the current practice of British critical care physicians in managing new-onset atrial fibrillation. The questionnaire was pilot-tested and revised based on the question value. Where clinically relevant, more than one answer could be given.

### Selection sample

The survey population consisted of 3,152 medical members of Intensive Care Society (ICS), who are practicing intensive care doctors in the UK and for whom email contacts were available. We excluded critical care doctors who did not practise or were retired.

The participants received an email containing the link to the questionnaire, which was a web-based survey comprising of 21 questions and developed using the open source survey application Qualtrics®. The online survey was conducted from November 2016 to December 2016.

## Questionnaire

The survey was anonymous and included a consent form at the beginning of the survey and in the email circulated to all participants. The consent form explained the purpose of the study, the risks and benefits and data management. The survey consisted of 21 questions, which were divided into two domains. The first domain comprised seven questions, which mainly recorded the background and demographic data of the responding physicians and their critical care unit. The second domain had 14 questions that aimed to identify current treatment strategies for critically ill patients with new-onset fast atrial fibrillation. The survey took no more than 10 min to complete.

Demographic variables included type of practice hospitals (district general, tertiary referral centre or university hospital), the number of patients admitted to the unit per year, the number of staffed beds in the department, the case mix of the unit department (predominantly surgical, medical, mixed or specialist ICU) as well as the level of training, years of experience and any secondary specialty of the survey participants.

To identify and explore the current practice in managing critically ill patients with new-onset AF, participants were asked to state a threshold heart rate at which they intervene in patients with fast AF and stable blood pressure, whether they favour rhythm or rate control as the primary treatment goal, the most commonly used anti-arrhythmic drugs and the reasons for their choice. Survey participants were also asked about their treatment strategy in an example of a critically ill patient with chest sepsis who developed new-onset AF with a heart rate of 140–160 beats per minute. In this survey question, participants were given options of different treatment strategies, including electrolyte supplementation, Direct Current (DC) cardioversion and anti-arrhythmic drugs. Survey participants were requested to state their target serum potassium and serum magnesium level among critical care patients with new-onset AF.

Finally, participants were also asked about their anti-coagulation practice including duration and medication used and their views on stroke risk assessments in critically ill patients with new-onset AF. A further question explored the use of transoesophageal or transthoracic echocardiography to guide treatment. At the end of the survey, their views on conducting a clinical trial investigating treatment of new-onset AF in this subgroup of patients were obtained. This included the anti-arrhythmic medications at highest priority for investigation and acceptability of a placebo arm in a research study investigating the effectiveness of anti-arrhythmics in critically ill patients with new-onset AF.

The full content of the survey is available as Supplemental Information 1.

## Data analysis

A survey data report was auto-generated by Qualtrics® to aid data analysis. Descriptive statistics were carried out by providing absolute numbers and percentages of background and demographic variables and for all questions relating to knowledge and current practice. Where applicable, contingency tables were produced and analysed using Fisher's exact test. To measure of association between two nominal variables, Cramer's V was used and interpreted as follows: 0.1–0.29 weak association, 0.3–0.5 moderate association, >0.5 strong association.

## RESULTS

### Demographics of survey participants

Questionnaires were sent to 3,152 members of the Intensive Care Society (ICS) who had updated email addresses available. We received 427 responses. Survey links that had been opened without provision of replies were excluded from the study. 397 complete survey responses were obtained (12.6%). We excluded 30 surveys with incomplete answers on the management of new-onset AF in critically ill patients from the analysis. Four lacked information on background and demographics and were also excluded.

A total of 46.6% of respondents were from District General Hospitals; a smaller proportion (38.5%) worked in University Hospitals or Tertiary referral centres. The admission rate ranged between 500 and 2,000 patients/year in most centres, 81.1% of units were mixed ICUs admitting medical and surgical patients (Table 1).

Most respondents (47.2%) worked on units with 11–20 staffed beds (level 2 and level 3). Our results indicate that mainly senior medical staff responded to the survey invitation, with Consultants representing 71.8% of respondents; 53.9% had more than 10 years of experience in Critical Care. Anaesthesia was the most common secondary specialty stated (83.9%) (Table 2).

### Anti-arrhythmic treatment of new-onset AF

A total of 39.5% of respondents would start intervention on patients with fast new-onset AF and stable blood pressure at a heart rate between 120 and 139 beats/min. However, a similar proportion of respondents (34.8%) would treat all patients who developed new-onset fast AF, independent of their heart rate, even if the blood pressure remained stable. A total of 54.7% of respondents stated that the primary treatment goal of new-onset AF among ICU patients with stable BP without a known cardiac history was rate control, while 40.3% of respondents aimed for rhythm control. We analysed whether choosing rate control versus rhythm control as primary treatment goal influenced physicians' views on heart rate at which they would intervene (Table 2). Although a significantly higher percentage ($p < 0.001$) of physicians who stated "Rhythm control" as their primary treatment goal would intervene at any heart rate, the overall association between primary treatment goal and heart rate requiring intervention was only moderate (Cramer's $V = 0.316$).

Amiodarone was by far the most commonly used anti-arrhythmic for treatment of new-onset AF in critically ill patients (80.9% of answers), followed by beta-blockade (11.6%) (Fig. 1).

We wanted to explore the primary treatment strategy for a typical critically ill patient using a case vignette of a patient with chest sepsis, who develops fast new-onset AF with a heart rate of 140–160 bpm. The patient had no cardiac history, a blood pressure of 100/60 mmHg and received 0.25 mcg/kg/min noradrenaline. Most survey respondents opted for electrolyte supplementation to high normal level and additional anti-arrhythmics (53.6%). A total of 23.8% of respondents advocated electrolyte supplementation to a high normal level only, and 14.9% opted for DC cardioversion when anti-arrhythmics and electrolyte replacement fail to achieve rate and/or rhythm control. No one chose to perform DC cardioversion only (Fig. 2). In critically ill patients with new-onset AF

**Table 1    Intensive Care Unit (ICU) characteristics and level of Level of training of survey respondents.**

**(A)**

| Which of the following answers most accurately describes your hospital? | Response | % |
|---|---|---|
| District General Hospital | 185 | 46.60% |
| Teaching Hospital | 59 | 14.86% |
| Tertiary Referral Centre or University Hospital | 153 | 38.54% |
| Total | 397 | 100.00% |

**(B)**

| How many patients are admitted to your Intensive Care Unit per year? | Response | % |
|---|---|---|
| <500 | 44 | 11.08% |
| 500–1000 | 140 | 35.26% |
| 1000–2000 | 129 | 32.49% |
| >2000 | 49 | 12.34% |
| I do not know | 35 | 8.82% |
| Total | 397 | 100.00% |

**(C)**

| Please state the case mix of your ICU/HDU: | Response | % |
|---|---|---|
| Predominantly surgical | 19 | 4.79% |
| Predominantly medical | 16 | 4.03% |
| Mixed ICU | 322 | 81.11% |
| Specialist ICU (please name specialty): | 40 | 10.08% |
| Total | 397 | 100.00% |

**(D)**

| Please state your level of training: | Response | % |
|---|---|---|
| Consultant | 285 | 71.79% |
| Trainee | 94 | 23.68% |
| SAS | 18 | 4.53% |
| Total | 397 | 100.00% |

**(E)**

| How many years of experience do you have in Critical Care? | Response | % |
|---|---|---|
| <1 year | 4 | 1.01% |
| 1–3 years | 42 | 10.58% |
| 3–5 years | 44 | 11.08% |
| 5–10 years | 93 | 23.43% |
| More than 10 years | 214 | 53.90% |
| Total | 397 | 100.00% |

**Table 1** (*continued*)

| (F) Please state your complimentary specialty: | Response | % |
|---|---|---|
| Intensive Care Medicine only | 37 | 9.32% |
| Anaesthesia | 333 | 83.88% |
| Acute medicine | 18 | 4.53% |
| Emergency Medicine | 6 | 1.51% |
| Paediatrics | 0 | 0.00% |
| Surgery | 3 | 0.76% |
| Total | 397 | 100.00% |

**Notes.**
SAS, Specialty and associate specialist (Doctors not in training with at least four years postgraduate experience).

**Table 2** Survey responses regarding heart rate at which doctors would intervene depending on their primary treatment goal (rate versus rhythm control).

| | | | Rate control | Rhythm control | Total |
|---|---|---|---|---|---|
| At which heart rate would you intervene in patients with fast AF and stable blood pressure? | >160/beats per min | Count | 3 | 0 | 3 |
| | | Percentage | 1.5% | 0.0% | 0.9% |
| | 100–119/beats per min | Count | 30 | 17 | 47 |
| | | Percentage | 15.2% | 11.5% | 13.6% |
| | 120–139/beats per min | Count | 93 | 48 | 141 |
| | | Percentage | 47.0% | 32.4% | 40.8% |
| | 140–159/beats per min | Count | 30 | 9 | 39 |
| | | Percentage | 15.2% | 6.1% | 11.3% |
| | Independent of their heart rate I treat all patients who have developed new onset fast AF even if the blood pressure remains stable | Count | 42 | 74 | 116 |
| | | Percentage | 21.2% | 50.0% | 33.5% |
| Total | | Count | 198 | 148 | 346 |
| | | Percentage | 100.0% | 100.0% | 100.0% |

74.3% of survey respondents aimed for serum Potassium level of >4.5 mmol/l and 59.7% of survey respondents aimed for serum Magnesium level of 1.0–1.2 mmol/l (Fig. 3). Pharmacodynamics properties and adverse effect profile were leading factors for choosing anti-arrhythmic treatment for new-onset AF in critically ill patients (Fig. 4).

## Anti-coagulation in patients with new-onset fast AF

A total of 63.8% of respondents stated that they would not regularly anti-coagulate critically ill patients with new-onset fast AF, while 30.8% would anti-coagulate within 72 h (Table 3A). A total of 53.3% of all survey respondents thought that subcutaneous low molecular weight heparin in therapeutic dose is appropriate for anti-coagulation and 26,8% considered intravenous high molecular weight heparin as appropriate, provided that no contra- indications for either substance were known (Table 3B). Sub-analysis excluding respondents who gave "I do not regularly anti-coagulate critically ill patients with new onset fast AF" as the only answer (136/362 respondents), revealed that either low or high

a)

| Which is the most commonly used anti-arrhythmic drug for new onset fast atrial fibrillation in your ICU? | Response | % |
|---|---|---|
| Amiodarone | 293 | 80.94% |
| ß-blocker | 42 | 11.60% |
| Flecainide | 1 | 0.28% |
| Diltiazem | 0 | 0.00% |
| Digoxin | 12 | 3.31% |
| Other anti-arrhythmics (e. g. Verapamil, sotalol) | 14 | 3.87% |
| Total | 362 | 100.00% |

b)

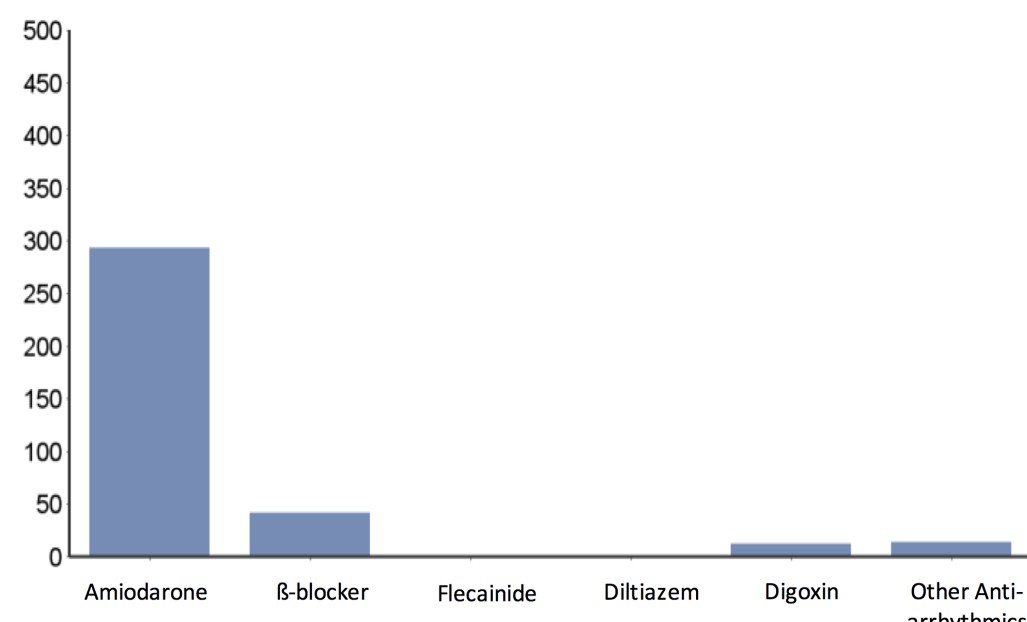

**Figure 1** **Medications used for treatment of new-onset atrial fibrillation.**

molecular weight heparin was considered appropriate for anti-coagulation by more than 98% of physicians.

A total of 68.0% of survey respondents did not use stroke risk scores routinely in critically ill patients with new-onset AF to assess the need for anti-coagulation. A total of 30.9% of survey respondents thought that stroke risk scores inaccurately reflect the risk of embolic events in critically ill patients with new-onset atrial fibrillation due to prothrombotic changes associated with critical illness. A total of 47.0% of respondents thought that

a)

| In a patient with chest sepsis (no cardiac history, blood pressure 100/60 mmHg, receiving 15 ml/min (0.25 mcg/kg/min) noradrenaline), who develops fast new onset AF with a heart rate of 140-160 bpm, your primary treatment strategy consists of: | Response | % |
|---|---|---|
| Supplement electrolytes (magnesium and/or potassium) to a high normal level | 86 | 23.76% |
| Supplement electrolytes to a high normal level and anti-arrhythmics | 194 | 53.59% |
| Supplement electrolytes to a high normal level and DC cardioversion | 23 | 6.35% |
| Anti-arrhythmics only | 3 | 0.83% |
| DC cardioversion only | 0 | 0.00% |
| DC cardioversion when anti-arrhythmics and electrolyte replacement fail to achieve rate and/or rhythm control | 54 | 14.92% |
| I only intervene if blood pressure drops or inotrope requirements increase | 2 | 0.55% |
| Total | 362 | 100.00% |

b)

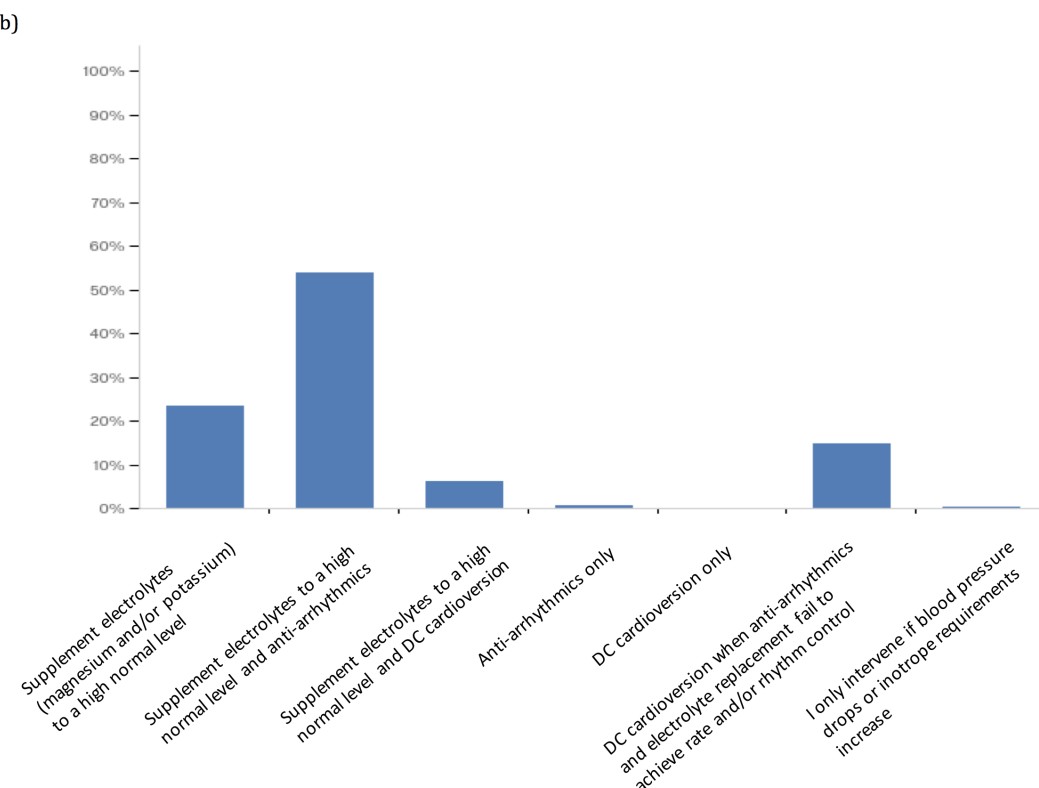

**Figure 2  Case vignette to assess treatment of new-onset atrial fibrillation with fast ventricular rate.**

modified risk scores should be developed for critically ill patients with new-onset atrial fibrillation (Table 4).

Approximately half of the respondents (52.2%) would request an echocardiogram in patients with new-onset AF. Only a small minority (3.3%) would use transoesophageal echocardiography, while a large proportion (48.9%) would request a transthoracic echocardiography. 39.2% of survey respondents did not routinely perform echocardiography to guide treatment.

a)

| In critically ill patients with new onset fast AF, which Serum Potassium level would you aim for? | Response | % |
|---|---|---|
| >3.5mmol/l | 3 | 0.83% |
| >4 mmol/l | 82 | 22.65% |
| >4.5 mmol/l | 269 | 74.31% |
| >5 mmol/l | 7 | 1.93% |
| I do not aim for a specific serum potassium level | 1 | 0.28% |
| Total | 362 | 100.00% |

b)

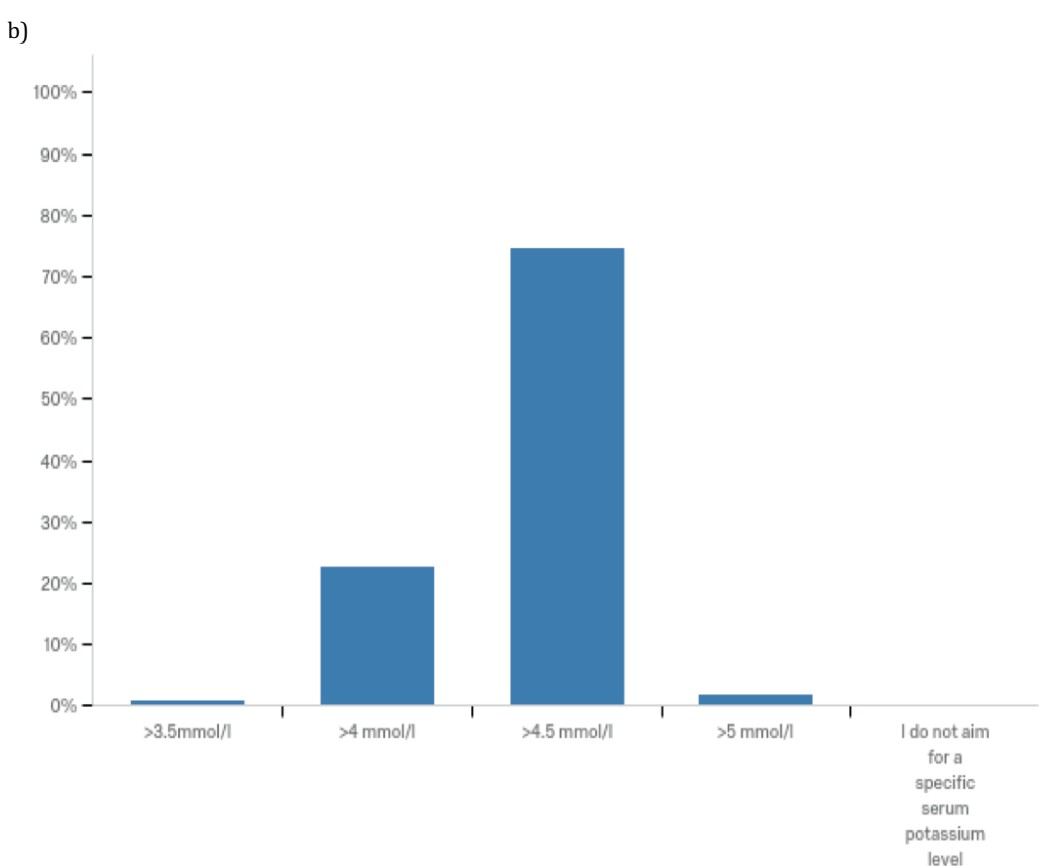

**Figure 3** **Electrolytes level targets in the management of atrial fibrillation.** (continued on next page...)

c)

| In critically ill patients with new onset fast AF, which Serum Magnesium level would you aim for? | Response | % |
| --- | --- | --- |
| 0.75-1 mmol/l | 30 | 8.29% |
| 1.0-1.2 mmol/l | 216 | 59.67% |
| >1.2 mmol/l | 71 | 19.61% |
| I do not aim for a specific serum magnesium level | 45 | 12.43% |
| Total | 362 | 100.00% |

d)

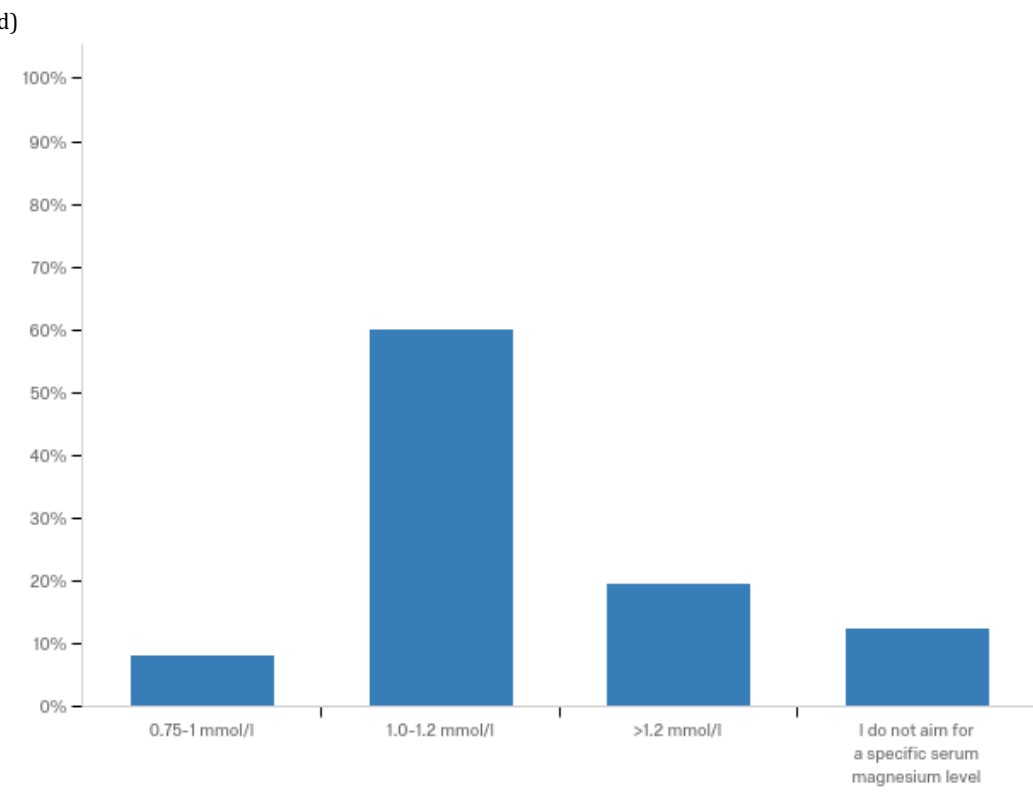

**Figure 3 (…continued)**

a)

| Rationale for choosing anti-arrhythmic treatment for new-onset AF in critically ill patients. | Response | % |
|---|---|---|
| Availability from hospital pharmacy | 53 | 8.40% |
| ICU drug policy | 109 | 17.27% |
| Cost | 12 | 1.90% |
| Pharmacokinetic advantages | 109 | 17.27% |
| Pharmacodynamic properties | 176 | 27.89% |
| Adverse effect profile | 123 | 19.49% |
| Other, please specify: | 49 | 7.77% |
| Total | 631 | 100% |

b)

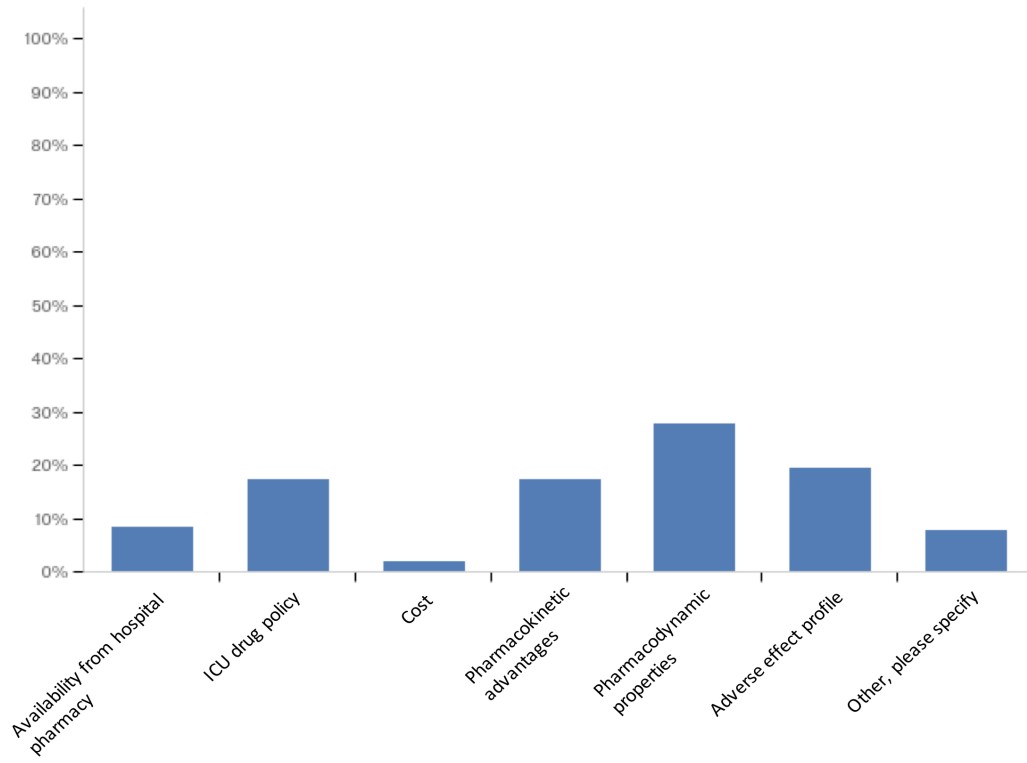

**Figure 4** **Rationale for choosing anti-arrhythmic treatment for new-onset AF in critically ill patients.** Other reasons given included "Effectiveness", "Chance of cardioversion", "Amiodarone works".

## Research interest amongst intensivists who treat patients with new-onset AF

A total of 85.4% of survey respondents would consider taking part in a clinical trial investigating treatment of new-onset fast AF in the critically ill. In the setting of a research study on general ICU patients, amiodarone (74.0%) and beta-blockers (55.8%) were the two most frequently mentioned pharmaceutical treatments to be investigated for critically

**Table 3** Survey responses regarding initiation of anti-coagulation treatment in atrial fibrillation and choice of appropriate anticoagulants.

**(A)**

| When would you normally anti-coagulate critically ill patients with new-onset atrial fibrillation, if no contra-indications for anti-coagulation are known? | Bar | Response | % |
|---|---|---|---|
| New onset AF within 24 h | | 26 | 7.18% |
| New onset AF within 48 h | | 54 | 14.92% |
| New onset AF within 72 h | | 35 | 9.67% |
| Before starting anti-arrhythmic medication | | 4 | 1.10% |
| After starting anti-arrhythmic medication | | 3 | 0.83% |
| Before DC cardioversion | | 9 | 2.49% |
| I do not regularly anti-coagulate critically ill patients with new-onset fast AF | | 231 | 63.81% |
| Total | | 362 | 100.00% |

**(B)**

| In critically ill patients with new-onset fast atrial fibrillation which of the following do you consider appropriate for anti-coagulation provided that no contra-indications are known? Please tick all answers that reflect your views | Bar | Response | % |
|---|---|---|---|
| Intravenous High Molecular Weight heparin in therapeutic dose | | 97 | 26.80% |
| Subcutaneous Low Molecular Weight heparin in therapeutic dose | | 193 | 53.31% |
| Use of novel oral anti-coagulants (NOACs) | | 17 | 4.70% |
| Use of warfarin | | 20 | 5.52% |
| I do not regularly anti-coagulate critically ill patients with new onset fast AF | | 190 | 52.49% |
| Total | | 517 | 100.00% |

**Table 4** Survey responses regarding stroke risk assessment in critically ill patients with new-onset atrial fibrillation.

| Please tick all answers that reflect your views on stroke risk assessment in critically ill patients with new-onset atrial fibrillation: | Bar | Response | % |
|---|---|---|---|
| I do not use stroke risk scores routinely in critically ill patients with new onset AF to assess the need for anti-coagulation | | 246 | 67.96% |
| I regularly calculate a risk score (e.g., CHAD2, CHA2DS2-VASc) to assess the need for anti-coagulation | | 39 | 10.77% |
| Stroke risk scores inaccurately reflect the risk of embolic events in critically ill patients with new-onset atrial fibrillation due to pro-thrombotic changes associated with critical illness | | 112 | 30.94% |
| Stroke risk scores favour anti-coagulation despite a higher risk of bleeding in critical illness | | 73 | 20.17% |
| Modified risk scores should be developed for critically ill patients with new-onset atrial fibrillation | | 170 | 46.96% |
| Total | | 640 | 100.00% |

ill patients with new-onset fast AF. There was a divided opinion regarding the use of a placebo arm (i.e., not treating new-onset AF with anti-arrhythmics or rate-limiting agents, when BP and cardiac output are stable). In the setting of a research study investigating the effectiveness of anti-arrhythmic or rate-limiting agents in critically ill patients with new-onset fast AF, 50.3% of respondents would not accept a placebo arm.

## DISCUSSION

New-onset AF is the most commonly observed arrhythmia in critically ill patients (*Artucio & Pereira, 1990*). It is associated with a worse prognosis (*Champion et al., 2014*; *Chen et al., 2015*; *Shaver et al., 2015*) and longer ICU stays (*Duby et al., 2017*; *Reinelt et al., 2001*), particularly in patients with sepsis, (*Klein Klouwenberg et al., 2017*). Despite the large number of patients affected and potential long-term consequences (*Walkey et al., 2014*), thorough research on treatment of new-onset AF is lacking (*Walkey, Hogarth & Lip, 2015*). Our survey revealed two areas with substantial research need: Anti-arrhythmic treatment of new-onset AF and anticoagulant therapy.

Prolonged periods of elevated heart rate are associated with a higher risk of cardiac complications (*Sander et al., 2005*). A retrospective study in non-cardiac ICU patients found that 37% of patients with new-onset AF suffered from hemodynamic instability related to the development of AF (*Kanji et al., 2012*). Intensivists are concerned about complications such as hypotension, myocardial ischemia and reduced organ perfusion, all of which are known to be associated with new-onset AF with a high conduction rate. In our survey, one third of intensivists would treat all patients with new-onset fast AF, independent of their heart rate, even if the blood pressure remained stable, presumably in an attempt to prevent these potentially harmful events. However, a slightly higher proportion (39.5%) would set a heart rate of more than 120 bpm as a treatment threshold in haemodynamically stable patients.

More than half of the intensivists who participated in our survey aimed for rate control as the primary treatment goal, while about 40% prioritized rhythm control. A recent single-center, retrospective, cohort study described a higher mortality in septic patients with new-onset AF who failed to convert into sinus rhythm (*Liu et al., 2016*). Rhythm control and rate control have both been used as outcome measures in the few therapeutic studies available in the literature, although there is no clear guidance whether rate or rhythm control is preferable to influence overall outcome of critical illness (*Liu et al., 2016*).

In fact, a recent systematic review on new-onset AF in non-cardiac critically ill patients identified only five studies, which compared different treatment strategies for new-onset AF in critically ill patients (*Yoshida et al., 2015*). However, only one of the studies was a randomised controlled trial. Therapies studied to date include beta-blockers (*Balser et al., 1998*; *Meierhenrich et al., 2010*), Vernakalant (*Arrigo, Bettex & Rudiger, 2014*), Diltiazem (*Balser et al., 1998*), DC cardioversion (*Arrigo et al., 2015*; *Kanji et al., 2012*; *Mayr et al., 2003*; *Meierhenrich et al., 2010*; *Seguin et al., 2006*), Amiodarone (*Balser et al., 1998*; *Kanji et al., 2012*; *Meierhenrich et al., 2010*; *Seguin et al., 2006*; *Sleeswijk et al., 2008*), Sotalol (*Kanji et al., 2008*) and digitalis glycosides (*Meierhenrich et al., 2010*; *Seguin et al., 2006*). The need

to perform research on treatment of new-onset AF in critically ill patients is reflected by the high percentage (85%) of intensivists who declared an interest in participating in such research.

In our survey, amiodarone was the preferred pharmaceutical treatment for more than 80% of intensivists who participated, potentially because it is associated with fewer haemodynamic effects compared to beta-blockers and calcium channel blockers (*Delle Karth et al., 2001*). Unfortunately, evidence supporting its use in critically ill patients is limited to small single-centre studies only (*Delle Karth et al., 2001*; *Shibata et al., 2016*).

NICE guidelines recommend either a standard beta-blocker or a rate-limiting calcium-channel blocker as initial monotherapy for patients with new-onset AF (*Jones et al., 2014*) or, for patients who are sedentary, digoxin monotherapy. These guidelines have been developed for patients with AF in the general population, but may not be transferable to the intensive care setting due to different predisposing factors for AF in critically ill patients. In particular, new-onset AF has been associated with inflammation and occurs in up to 46% of patients with septic shock (11). Patients with septic shock who developed new-onset AF showed a continuous, significant increase in CRP plasma levels before occurrence of AF (*Meierhenrich et al., 2010*). Further risk factors for new-onset AF in the critically ill include inotropic support (*Seguin et al., 2006*), advanced age and high scores of severity of disease (*Yoshida et al., 2015*). In the United Stated, where diltiazem is available for intravenous administration, calcium-channel blockers were the most frequently used drugs to treat new-onset AF during sepsis (*Walkey et al., 2016*), although the use of beta-blockers was associated with improved mortality in a propensity analysis.

About a quarter of respondents would choose electrolyte supplementation to a high normal level only, while more than half (53.6%) would add anti-arrhythmics or rate-limiting agents as a primary treatment strategy. The use of intravenous magnesium sulphate bolus application followed by continuous infusion achieved conversion to sinus rhythm or decrease in heart rate <110 bpm in 16 of 29 patients (55%) in a small prospective study (*Sleeswijk et al., 2008*), while magnesium-amiodarone step-up therapy achieved a conversion rate of more than 90% within 24 h in a cohort of mixed critically ill patients. Although these findings suggest that amiodarone, magnesium sulphate, or the combination thereof might be effective to prevent or treat new-onset AF, more studies are needed to evaluate the efficiency and safety profile of these drugs in the general critically ill population.

Haemodynamic stability during and immediately after the onset of AF and long-term stroke risk are often dependent on underlying left ventricular systolic and diastolic function. Echocardiography provides useful information on right and left ventricular function as well as size of both atria to determine optimal treatment in patients with acute haemodynamic deterioration. Although nearly half of the respondents would request a transthoracic echo, a large proportion (39%) does not routinely use echocardiography to guide management in this patient cohort.

NICE/ACC/AHA/ESC practice guidelines recommend routine anti-coagulation of patients with new-onset AF depending on their individual risk of thromboembolic events using established scores such as CHADS2 and CHA2DS2-VASc (*Jones et al., 2014*). However, the risk of stroke and thromboembolic events in critically ill patients who

develop new-onset AF has been evaluated in only very few studies. A recent prospective observational study revealed that both CHADS2 and CHA2DS2-VASc are predictive of thromboembolic events in the critical care setting (*Champion et al., 2014*), with a CHADS2 score of 4 or higher being the most accurate threshold. In a large retrospective study on more than 49,000 patients with sepsis, *Walkey et al. (2011)* described that new-onset AF during severe sepsis was associated with a nearly four-fold increased risk of in-hospital ischemic stroke, with threefold greater in-hospital stroke rates compared with patients without AF during sepsis. Despite this adverse risk profile, 63.8% of intensivists participating in our survey stated that they would not regularly anti-coagulate critically ill patients with new-onset fast AF, while 30.8% of respondents would anti-coagulate within 72 h. Intensivists may be reluctant to commence therapeutic anti-coagulation in critically ill patients with new-onset AF because the risk/benefit ratio of anti-coagulation during acute critical illness is often unclear. Critically ill patients may be at substantially increased risk of severe bleeding due to thrombocytopenia, renal failure, liver failure, invasive devices, and unscheduled procedures (*Walkey, Hogarth & Lip, 2015*), and bleeding complications such as gastrointestinal haemorrhage or intracerebral bleeds are common (*Darwish et al., 2013*). Currently recommended scores for stratification of thromboembolic risk, such as the CHADS2 and CHA2DS2VASc scores, and the scores for hemorrhagic risk, like the HAS-BLED score have limitations when applied in critically ill patients (*Ferreira et al., 2015*). Hence scores developed to assess the stroke risk in the general population with AF, are not routinely used by more than two thirds of intensivists who participated in our survey. 30.9% of survey respondents felt that the currently recommended stroke risk scores inaccurately reflect the risk of embolic events in critically ill patients with new-onset AF due to prothrombotic changes associated with critical illness. Nearly half of the respondents (47.0%) thought that modified risk scores should be developed for critically ill patients with new-onset atrial fibrillation to take into account such alterations in the coagulation system in critically ill patients. In an attempt to address this, a multi-centre observational study has been set up to further identify clinical and echocardiographic risk factors for thromboembolic events in critically ill patients with new-onset AF (*Labbe et al., 2015*).

Importantly, decisions to anti-coagulate patients with new-onset AF during critical illness may influence the stroke risk beyond their critical care admission. More than half of patients with new-onset AF have a later recurrence of AF (*Walkey et al., 2014*). Patients with new-onset AF during acute illness also have an elevated long-term risk of stroke. A proposed approach to long-term management of patients who develop periods of AF during critical illness includes re-assessment for rhythm and heart rate surveillance, cardiovascular comorbidities, thyroid function, stroke risk and should include echocardiography and patients' preferences (*Walkey, Hogarth & Lip, 2015*) to manage stroke and cardiovascular risk after ICU discharge. However, data are lacking to estimate rates of severe bleeding versus stroke risk with use of systemic anti-coagulation during critical illness. Due to this uncertainty, a recent review concluded that routine anti-coagulation for new-onset AF among critically ill patients with elevated bleeding risk cannot be recommended as a treatment where benefits outweigh risks (*Walkey, Hogarth & Lip, 2015*).

Our survey has several limitations. We only approached members of the Intensive Care Society (ICS), due the necessity of an accessible and up-to-date email distribution list. Approximately two thirds of practising intensivists in the UK are members of the ICS, indicating that a significant proportion of intensivists were not given the opportunity to express their views. Given the low overall response rate, non-response bias may also have influenced our results. However, acute care physicians are a professional group with reportedly low survey response rates (*Champion & Deye, 2017*), likely due to time constraints and high workload. This short online survey consisted of a limited number of general questions only to allow rapid completion. Hence, we could not thoroughly explore background knowledge and experience of the respondents in in this complex field. In particular, treatment strategies in relation to cardiac function and combination of therapies such as cardioversion in addition to anti-arrhythmic substances could not be evaluated.

Our survey focused on general management of new-onset atrial fibrillation in critical care; specific subgroups of patients including those after cardiothoracic surgery or with primary cardiac diagnoses were not specifically addressed. These patient cohorts may require separate investigations because of different pathomechanisms and treatment requirements.

## CONCLUSIONS

We identified a high variation in the use of anti-arrhythmic treatment in critically ill patients with new-onset AF, which can be explained by the lack of both high-quality studies and evidence to guide the management of new-onset AF in the critical care setting. Intensivists expressed a substantial interest in research in this area, making it a priority for future clinical trials. Interventional and observational studies will have to address the benefits of pharmacotherapy for new-onset AF in critically ill patients and the effect of individual drug choices on patient short and long-term outcomes. Further work is also required to describe and assess the risk/benefit ratio of thromboembolic prophylaxis in critically ill patients with new-onset AF.

## ACKNOWLEDGEMENTS

We would like to thank Dr. Gary Masterson and Helle Sorensen for facilitating distribution of the questionnaire and John Jones and Anna Ripley for providing data about ICS and FICM membership. We would also like to thank all members of the Intensive Care Society who took the time to respond to the survey invitation.

### Funding

The authors received no funding for this work.

### Competing Interests

The authors declare there are no competing interests.

## Author Contributions

- Chung Shen Chean and Ingeborg Dorothea Welters conceived and designed the experiments, performed the experiments, analyzed the data, contributed reagents/materials/analysis tools, wrote the paper, prepared figures and/or tables.
- Daniel McAuley conceived and designed the experiments, contributed reagents/materials/analysis tools, reviewed drafts of the paper.
- Anthony Gordon contributed reagents/materials/analysis tools, reviewed drafts of the paper.

## Data Availability

   The raw data has been provided as a Supplementary File.

## Supplemental Information

Supplemental information for this article can be found online at http://dx.doi.org/10.7717/peerj.3716#supplemental-information.

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
