# Peer review of "Current practice in the management of new-onset atrial fibrillation in critically ill patients: a UK-wide survey"

_PeerJ, doi:10.7717/peerj.3716_

## Round 0.1 · original submission · Minor Revisions

My additional comments are:

1. Beta-blockers are rate slowing agents with the exception of Sotalol. The confusion in the text and abstract needs rectification. Please also rectify figure 1 accordingly.
2. Tables 1 and 2 should be combined
3. I do not see any added value in showing table 3 - wherby info can be conveyed efficiently in the text.
4. Figures 2-4: Y axis would be better in percentage
5. Figure 4 - not all texts are showing for he different bars? A table might do a better job.
6. There should be discussion regarding handover of patients for AF care post ICU stay - e.g. medical or cardiology unit referral - that may improve longer term outcome
7. This sentence needs rewriting: "Most survey respondents opted for electrolyte supplementation to high normal level and anti-arrhythmics (53.6%) to treat of a patient with chest sepsis, but no cardiac history, a blood pressure of 100/60 mmHg and receiving 0.25 mcg/kg/min noradrenaline, who develops fast new-onset AF with a heart rate of 140-160 bpm. "
8. The CV stability and subsequent stroke risk is often dependent on underlying LV systolic function. More discussion on this is recommended.

Reviewer 1 ·

Basic reporting

No issues with basic reporting.

Experimental design

No issues with experimental design.

Validity of the findings

No issues with validity of findings.

Additional comments

I had the pleasure of reviewing the article submitted by Dr Chen and colleagues to PeerJ, entitled “Current practice in the management of new onset atrial fibrillation in critically ill patients: A UK wide survey”. In this article, Dr colleagues describe the results of a survey directed at members of the Intensive Care Society on the current practice and management of critically ill patients with AF. They received responses from 397 individuals (12.6% response rate). They identified differences in rate and rhythm control, and anticoagulation management, amongst responders. These findings highlight the uncertainty regarding how best to manage this common condition and, in support of this, the majority of responders would consider taking part in clinical trials in this area.

This is a useful, short succinct paper that draws attention to a common complication in clinical practice. Limitations of the paper include the restriction to UK-based intensivists and relatively low response rate.

Reviewer 2 ·

Basic reporting

No comment

Experimental design

No comment

Validity of the findings

No comment

Additional comments

Minor comments:
- Line 255: An important study adressing DC-Cardioversion is not cited: Arrigo et al, Crit Care Med 2015. Please add it to the references.
- Line 255: Other substances were studied for treating AF in the ICU setting (e.g. esmolol, vernakalant). Please consider to cite these studies: Mooss et al. Am Heart J 2000. Rudiger et al. Crit Care Res Pract 2014)

- Some typos:
-- line 163 (38.5%): brackets are lacking;
-- lines 199 and 201: "Low Molecular Heparin" and "High Molecular Heparin": no capital letters are required
-- line 214: "Echocardiography": no capital letters are required

·

Basic reporting

1- The article is written in clear, unambiguous, professional English language. It is well referenced; however gathering the large amount of literature referred should be envisaged.

Figures are relevant, but should be better labelled & described (a)b)c) are not correctly placed; is Figure 4 really useful?).

Raw data are supplied.

Experimental design

Research question is well defined, relevant & meaningful. It is stated how the research fills an identified knowledge gap. Investigation is performed to a high ethical standard. Please consider reporting the data using RAND-UCLA methodology for improved clarity and relevance.

Methods are described with sufficient detail & information to replicate.

Validity of the findings

3- Impact and novelty are assessed. It is the first survey about AF in the ICU where most decisions are empirical. Thus, this survey may impact clinicians’ decisions.

Conclusions are well stated, linked to original research question & limited to supporting results.

Additional comments

4- Dr CHEN and colleagues report a timely survey about a very common, serious issue in critically ill patients: atrial fibrillation (AF). As well described all long their article, AF is a frequent and major complication of ICU stay but very few trials assessed its treatment regarding anti arrhythmic and anticoagulant drugs. Thus a survey is up-to-date and may impact clinicians’ decisions.
International Guidelines (reference 26 cited, consider citing others including Levy B AIC 2015 which is specifically dedicated to critically ill patients with cardiogenic shock) give a framework for the treatment of AF, including for those patients in the ICU: direct current cardioversion is strongly advised for poorly tolerated AF.

A major limitation of their study is the only small place for respondents to embrace such a large and complex problem: only 14 questions may not allow delineating multiples facets of new onset AF or specific ICU issues. Accordingly, there is no differentiation of anti arrhythmic drugs based on structural cardiopathy or heart failure, no notion of drug-enhancement cardioversion (ref 26)… As every online survey using this methodology, such limitations and low response rate occur (consider citing reference: Champion S. and Deye N. 2017). Another limitation that should be added in the limitations is the lack of knowledge of the respondents (mostly anaesthesiologists) in a very complex field.

Reporting the response rates are confusing in Table 3 and Lines 172 and 241: if 39.5% would start intervention at HR 120-139, one should read that 173/362 (47.79%) would start any intervention for fast AF, 316/362 (87.29%) if HR>120/min and so on. Accordingly, the authors correctly made the addition for Table 4, line 198.

Moreover, the authors should report the results depending on the respondents’ “philosophy” : rhythm or rate control. Thus in Table 3b), 146 respondents (40.33%) reported a rhythm control strategy. We can assume that mostly belong to the group that start an intervention whatever heart rate is (n=126), or for fast AF <120/min. Accordingly, respondents with rate control “philosophy” may be less prone to intervene. Furthermore, Table 4b) is hard to interpret in that way: from 517 answers, were there respondents who gave both answers “I do not regularly” and “I consider heparin appropriate”? Splitting the results may add information.

Apart from those issues that should be corrected before Acceptance, only minor issues appear.

Please consider separating the concepts of “new onset” and “fast” AF, as it refers to two different subjects, line 75.
Line 117: please delete “and.”
Line 135: please spell DC for first use (instead of line 255).
Line 162: please use the “majority” as appropriate.

Discussion: please do not repeat the introduction: in this context, references to independent adverse prognosis of AF should be gathered and only be briefly explained (higher ICU mortality: please add references to Meierheinrich ref 18 and Champion ref 28. The latter reference can further illustrate AF-induced hemodynamic compromise by increased catecholamine support in 53% or pulmonary oedema in 43%). Please be concise and consider discarding line 233, ref 15, and from line 269, “In particular… to line 273, … disease. Alternatively, if really considered major, risk factors for new onset AF should be briefly presented in the introduction.
Line 324, please separate “risks of stroke”.

Figure 1: verapamil and diltiazem should be part of the same ligne, as their share pharmacologic properties of non-dihydropyridin calcium channel blockers.
Table 2a), please explain SAS.

---

## Round 0.2 · accepted · Accept

Thank you for responding to the majority of the reviewers' and editor's comments.